# Alzheimer’s-Like Pathology at the Crossroads of HIV-Associated Neurological Disorders

**DOI:** 10.3390/vaccines9080930

**Published:** 2021-08-21

**Authors:** Divya T. Chemparthy, Muthukumar Kannan, Lila Gordon, Shilpa Buch, Susmita Sil

**Affiliations:** University of Nebraska Medical Center, Department of Pharmacology and Experimental Neuroscience, Omaha, NE 68198-5880, USA; divya.thomas@unmc.edu (D.T.C.); mkannan@unmc.edu (M.K.); lilagordon901@gmail.com (L.G.)

**Keywords:** HAND, AD, tauopathy, Aβ42, synaptodendritic alterations

## Abstract

Despite the widespread success of combined antiretroviral therapy (cART) in suppressing viremia, the prevalence of human immunodeficiency virus (HIV)-associated neurological disorders (HAND) and associated comorbidities such as Alzheimer’s disease (AD)-like symptomatology is higher among people living with HIV. The pathophysiology of observed deficits in HAND is well understood. However, it has been suggested that it is exacerbated by aging. Epidemiological studies have suggested comparable concentrations of the toxic amyloid protein, amyloid-β42 (Aβ42), in the cerebrospinal fluid (CSF) of HAND patients and in the brains of patients with dementia of the Alzheimer’s type. Apart from abnormal amyloid-β (Aβ) metabolism in AD, a better understanding of the role of similar pathophysiologic processes in HAND could be of substantial value. The pathogenesis of HAND involves either the direct effects of the virus or the effect of viral proteins, such as Tat, Gp120, or Nef, as well as the effects of antiretrovirals on amyloid metabolism and tauopathy, leading, in turn, to synaptodendritic alterations and neuroinflammatory milieu in the brain. Additionally, there is a lack of knowledge regarding the causative or bystander role of Alzheimer’s-like pathology in HAND, which is a barrier to the development of therapeutics for HAND. This review attempts to highlight the cause–effect relationship of Alzheimer’s-like pathology with HAND, attempting to dissect the role of HIV-1, HIV viral proteins, and antiretrovirals in patient samples, animal models, and cell culture model systems. Biomarkers associated with Alzheimer’s-like pathology can serve as a tool to assess the neuronal injury in the brain and the associated cognitive deficits. Understanding the factors contributing to the AD-like pathology associated with HAND could set the stage for the future development of therapeutics aimed at abrogating the disease process.

## 1. Introduction

Human immunodeficiency virus (HIV) is a retrovirus that has affected more than 40 million people worldwide. The increased life expectancy of HIV patients undergoing antiretroviral therapy has led to the emergence of other comorbid medical complications [1]. Neurocognitive dysfunction is detected in almost 50% of HIV patients, despite the use of combined antiretroviral therapy (cART), with increased incidence in older people [1,2,3,4]. Several studies have demonstrated the accumulation of toxic amyloid isoforms in the brains of HIV-1 patients on cART, resulting in increased numbers of HIV-affected people living with HIV (PLWH) with the comorbidity of Alzheimer’s disease (AD) with increasing age. Regarding the development of AD in older HIV survivors, however, it remains unclear whether it develops independently or as a comorbidity associated with HIV [4,5,6,7].

Among HIV-affected individuals with Alzheimer’s-like pathology [8,9], the infection likely modulates the pathological features associated with AD. For example, there is increased deposition of amyloid β (Aβ) plaques in the brains of HIV-affected individuals, and this is often seen as perivascular aggregates, which are present both as plaques and as intraneuronal inclusions: this is distinct from AD pathology, where the amyloid depositions are largely parenchymal and extracellular in nature [4,10]. The deposition of neurofibrillary tangles and amyloid plaques is considered the major hallmark of AD, and their role in disease pathogenesis remains an area of intense investigation. In the HIV-affected brain, the processes are even more complicated. In HAND, several mechanisms are involved, including oxidative stress and chronic microglial activation, leading to neuroinflammation, Aβ deposition, hyperphosphorylation of Tau protein, and the toxic effects of a long-term dependence on cART [11]. It has been shown that certain antiretroviral medications, particularly reverse transcriptase inhibitors, could have additive amyloidogenic effects on macrophages. However, these effects were observed at very high concentrations that are unattainable in the brains of HIV-affected individuals [12]. Even though cART successfully suppresses viral replication, HIV regulatory proteins such as trans-activator of transcription (Tat), envelope glycoprotein (Gp120), viral protein R (Vpr) as well as a negative factor (Nef), which persist in the brains of suppressed individuals, can directly influence the CNS and activate the neuroinflammatory pathways, followed by neuronal injury and dysfunction. Several studies have reported the presence of these HIV proteins in the post-mortem brain samples of HIV-affected individuals. An earlier study has reported an estimated range of 100–120 µg/mL (wet weight of gray matter) of Tat protein in brain autopsy samples from HIV patients [13]. In another study, gp120 immunoreactivity was detected in the post-mortem brain tissue of all patients with HIV encephalitis [14]. Interestingly, the presence of Nef protein-positive astrocytes was reported in the brain tissue of HIV patients who had suffered from moderate to severe dementia [15]. Tat protein, which is produced as an early protein from the proviral DNA [16] from HIV reservoirs in the brain, is likely a major contributor to Alzheimer’s-like pathology. The linkage and causative mechanisms between neuroinflammation, HIV-CNS neuroinfections, HAND, and AD are not completely understood. It is thus important to understand the fundamental molecular linkages among these pathologies, which could shed light on the mechanisms involved in the disease pathogenesis and aid in the future development of therapeutics targeting the disease’s progression, while also facilitating disease diagnosis and prognosis. This review focuses on the cause–effect relationship of Alzheimer’s-like pathology in HAND, and the contribution of several CNS cell types to this process.

## 2. HIV-Associated Neurocognitive Disorder (HAND)

Though immune dysfunction and dysregulation are considered the primary contributors to HAND, cognitive impairments are the ultimate long-term consequence of HIV-1 infection. Approximately 30–50% of HIV-affected individuals are afflicted with HAND [17]. While the advent of cART has dramatically decreased the severity of HAND, the incidence of milder forms of HAND (asymptomatic and mild neurocognitive disorder) is on the rise and is expected to increase to 70% among aged HIV-affected individuals by 2030 [18]. Since the development of HAND affects day-to-day functioning, it is a topic of high clinical significance in PLWH [19]. Despite extensive research on HAND pathology, the lack of key clinical biomarker (s) and/or specific treatment (s) for the disease underscore the necessity for detailed research in this field.

## 3. Epidemiology

Since the commencement of the HIV-1 epidemic in the 1980s, conditions ranging from subtle neuropsychological impairments to intensely debilitating HAND have been frequently reported in HIV-affected individuals. The introduction of cART in 1996 dramatically improved both the longevity and neurological outcomes of patients. However, as the suppressed individuals continue to live longer and consequently consume cART medications over long periods of their lives, adverse neuropsychiatric effects of long-term cART are starting to become evident [2]. Several clinical studies have documented the persistence of mild-to-moderate neurocognitive impairment in PLWH receiving cART [4,20]. Indeed, cART initiation time also affects the severity of HAND. For example, it was reported that early cART initiation had better protective effects on HAND-like symptomatology [21]. In line with this study, another study also showed that the administration of cART immediately after HIV infection resulted in better protection from HAND, compared to delayed cART treatment 24 weeks after the onset of infection [22]. Often, asymptomatic neurocognitive impairments manifest as mild cognitive deficits that progressively transform into more severe forms of HAND. An assessment of HAND in a group of 197 HIV-affected individuals for a period of 6 years demonstrated an increased prevalence of HAND from 25–31% [23]. In fact, a recent study of the HIV-affected adult population from 32 countries from 1996 to 2020 reported an overall 42.6% prevalence of HAND (95% confidence interval [CI] 39.7–45.5). Worldwide, it is estimated that there are ~16,145,400 (95% CI 15,046,300–17,244,500) cases of HAND among HIV-affected adults [24].

## 4. Neuropathogenesis and Symptomology

It has been shown that a lack of or unsuccessful response to antiretroviral therapy results in HIV encephalitis and HIV leukoencephalopathy. The introduction of cART, however, has considerably changed the neuropathology of HAND, where the frequency of the severe form, HIV encephalitis, has been found to be significantly reduced from 54–15% [25]. The peripheral targets of HIV virions include circulating T-lymphocytes (CD4+) and macrophages [26]. Glycoprotein-mediated interactions between the host CD4 surface protein and C-C chemokine receptor type 5 (CCR5) facilitate the entry of the virus into the host [27]. Following infection, HIV-1 enters the CNS within days by crossing the blood–brain barrier (BBB) via the migrating monocyte and lymphocyte conduits. The presence of viral RNA has been detected in the cerebrospinal fluid (CSF) of patients within 8 days of initial infection [28]. The presence of perivascular macrophages expressing neurotoxic mediators generated from infected monocytes induces BBB permeability and facilitates virus entry into the brain [29,30].

A “trojan horse” mechanism of the entry of virus-infected CD4+ T-lymphocytes into the brain to further infect macrophages and microglia has been proposed [31]. It has been reported that the HIV virus not only transfers between infected CD4+ T cells and brain macrophages/microglia, but also between infected and non-infected macrophages [32]. Studies in SIV-infected macaques and HIV-affected individuals have shown that the interaction of non-infected macrophages with infected lymphocytes and macrophages produces virus-productive multinucleated giant cells [33,34,35]. The inaccessibility, at least in part, of cART in the brain macrophages and microglia makes them an established viral reservoir. The detection of low levels of HIV RNA in the CSF of patients even after 10 years of suppressive antiretroviral therapy (ART) confirms the brain as the key viral reservoir [36].

The expression of CD4 and CCR5 on microglia facilitates virus infection in an analogous manner to that observed in leukocytes [37]. Based on the premise that microglia play a key role in several neurodevelopmental functions, and since perivascular macrophages comprise important components of the immune responses in the brain, infected macrophages, and microglia are implicated in driving the cellular dysfunction underlying HAND. The secretion of cytokines and immune mediators from the infected cells is critical for the macrophage/microglia infection to progress into neurocognitive impairment [38]. Post-mortem studies on brain tissue from HIV-affected patients are suggestive of the presence of viral DNA and viral proteins in the astrocytes that lack CD4 expression, thereby highlighting a CD4-independent mechanism of virus infection [39,40]. Furthermore, the expression of human mannose receptor (hMR) was found to be crucial for the CD4-independent infectivity of HIV-1 in astrocytes [41]. Additional CNS cells that are vulnerable to HIV infection include pericytes, which are known to express high levels of CCR5 and low levels of CD4 protein. Pericytes play a key role in the formation and stabilization of the blood–brain barrier (BBB). Studies have shown that pericytes can also be the target of HIV-1 infection in the brain; the mechanisms of virus entry into these cells, however, need further investigation [42].

Despite the virus infecting the brain within days of systemic infection, the onset of HAND only manifests years post-infection [43]. Host-protective immunity eliminates and suppresses viral replication during the early course of viremia. However, as the disease progresses, owing to the failure of the host to mount antiviral responses, there is enhanced trafficking of infected cells into the CNS, resulting in the spread of viral species in the brain, ultimately leading to neurocognitive decline [43]. It must be noted that persistent low-level HIV infection also contributes to HAND [44]. Clinical studies in HIV patients on cART have identified two major sources of plasma viremia: the short-lived, actively infected CD4+T cells, and long-lived cells with the potential to induce viremia that is below the detection limit [45]. Although cART effectively suppresses viral replication, the persistence of the virus in long-lived resting immune cells such as myeloid cells lead to virus rebound upon cessation of therapy.

Interestingly, a recent longitudinal study revealed continued viral RNA decay with an average annual decrease by 6% in the HIV RNA level in the plasma of HIV patients on long-term ART [46]. Analysis of autopsy tissue from HIV patients on cART therapy revealed the persistence of both viral DNA and RNA, which was associated with pathological changes in the brain, lung, spleen, liver, lymph node, kidney, and aorta [47]. In another study of a cohort of 60 patients, it was shown that cART therapy with higher intracerebral penetration efficacy successfully suppressed HIV RNA viral load in both plasma as well as CSF [48]. In contrast, findings on patients with neurological symptoms found evidence for HIV resistance in the CSF, suggesting that persistent viral replication likely leads to neurological manifestations despite the suppression of plasma viremia with cART [49]. These studies thus underscore the association of HIV persistence with cognitive decline in a subset of patients.

There is a substantial change in the clinical manifestation of HAND over time. Comparative studies involving HIV-positive and HIV-negative subjects from the pre- and post-cART eras have shown that the pattern of neurocognitive impairments significantly differs between these periods, as impairments in motor skills, verbal fluency, and cognitive speed were primarily evident in the pre-cART patients, whereas memory and executive functioning impairments were more prevalent in the post-cART patients [2,50,51,52]. Impairments in prospective memory (“remembering to remember”) were found as a subtle cognitive change in HIV-affected individuals in the post-cART era, which could impact occupational functioning as well as medication adherence [53]. Figure 1 presents the clinical signs and symptoms of HAND. It is important to note that the clinical indices of cognitive dysfunction associated with HAND are more similar to neurodegenerative disorders such as AD. Amyloid deposition in the brain parenchyma with pathological levels of Aβ-42 indicates an association of AD-like pathology with HIV-associated neurological disorders (HAND) [54,55].

## 5. Alzheimer’s-Like Pathology of HAND

HIV infection primarily affects the immune system and significantly affects the brain [38]. The development of HIV pathogenesis substantially affects brain functioning, including memory, decision-making, movement skills, and behavior, often characterized as HAND [56]. Approximately 50% of PLWH on cART experience milder cognitive impairment, such as memory impairment and intellectual disability [57].

Over the past few years, our laboratory and many others have focused on the molecular mechanisms underlying HAND. The accumulation of insoluble amyloid beta-protein (Aβ) in the brain is one of the causative agents for HAND [26]. In AD, most of the Aβ protein exists as insoluble fibrils and neurofibrillary tangles [58,59]. While Aβ proteins are generally formed in the healthy brain throughout life, it is the insoluble form of Aβ proteins that accumulates in HAND or AD patients [8,9,54,60,61]. The sequential cleavage of amyloid precursor protein by β-secretase and γ-secretase produces Aβ to its monomeric form, with different lengths of peptides and monomeric amyloid proteins such as Aβ1-40 or Aβ1-42. In the healthy brain, the ratio of Aβ1-40 and Aβ1-42 is 90:10 and is present throughout life [59]. In AD, the monomeric form of Aβ1-40 or Aβ1-42 is converted into the toxic oligomeric and fibril forms and is a hallmark feature of AD. The accumulation of insoluble and oligomeric forms of Aβ contributes to cognitive impairment, leading to AD [62,63,64].

Additionally, amyloid plaques have also been demonstrated in the brains of patients suffering from HAND [65,66]. Amyloid plaques also accumulate in the CSF and blood. Recently, it was shown that HIV-Tat protein directly binds to the exterior surfaces of the Aβ fibrils, leading to increased β-sheet formation and lateral aggregation, consequently generating fibers with increased rigidity [67].

In clinical practice, the diagnosis of HAND can be achieved by amyloid positron emission tomography (PET) imaging, as well as by biochemical analysis of blood/plasma and CSF. Examining cognitive impairment by non-invasive methods using CSF and blood has the advantage of monitoring brain health in real-time [66].

Both HAND and AD are difficult to diagnose without quantitative neurocognitive testing, which is the most suitable means and the gold standard to study neurocognitive impairment. These procedures include the neuropsychological speech, attention, working memory, executive function, learning, recall, and motor functions [68].

## 6. Brain

Imaging-based techniques such as PET imaging [69,70,71] and structural magnetic resonance imaging (MRI) can aid in the clinical diagnosis of HAND. PET brain imaging is currently used in clinical and research settings and enables the in vivo detection of brain pathology in several neurological diseases, including HAND [72]. In addition to PET imaging, dynamic contrast-enhanced (DCE) MRI [71] has also been used to assess brain structure abnormalities. Neuroimaging can be used to determine the severity of brain injury and disease progression over time. The pathogenesis of HAND is likely to be multifactorial; however, evidence suggests that brain microglial activation is most likely a key pathogenic mechanism underlying the disease. PET imaging using radiotracers enables the study of the status of neuroinflammation and the functionality of the brain in order to understand the pathophysiology of HAND [73].

In addition to amyloid deposition and neuroinflammation, BBB integrity can also be assessed by MRI imaging [71,74]. Neuropathological examination of HIV patients revealed distinct multifocal-disseminated and diffuse brain tissue lesions, suggesting compromised BBB integrity in HIV+ patients [75]. In these patients, the severity of neurocognitive impairment correlated with the degree of BBB breakdown. Notably, the integrity of the BBB is affected in many diseases, such as AD, multiple sclerosis, and brain tumors [76,77]. In HAND, microvascular abnormalities have been observed in basal ganglia and the white matter of individuals by MRI imaging. Clinical and biochemical studies suggest that a disrupted BBB escalates the pathogenesis of HAND [78].

PET and MRI-based imaging techniques provide insights into the pathophysiological changes in the brain. However, a combination of imaging and biochemical analysis in CSF or blood is needed in order to provide a better understanding of HIV-associated neurocognitive diseases.

## 7. CSF

In addition to Aβ, phosphorylated-tau (p-tau) proteins are also associated with AD, particularly with alterations in their levels in the CSF of Alzheimer’s patients. Tau proteins are neuronal microtubule-associated proteins formed by alternative mRNA splicing, and they accumulate as neurofibrillary tangles in the brains of Alzheimer’s patients [79].

A study conducted by the Achim lab analyzed the levels of Aβ and p-tau proteins in post-mortem CSF collected from a group of well-characterized HIV-positive patients [80]. The levels of Aβ1-42 and Aβ1-40 in CSF were found to be reduced in HIV with HIV-associated dementia (HAD) patients but increased in those exposed to ART. Remarkably, the levels of p-tau proteins were inversely related to the expression of Aβ1-42 and Aβ1-40 in both HIV+ with or without ART cases [80].

Another study conducted by Gisslén et al. reported that the levels of amyloid precursor proteins alpha and beta (sAPPalpha and sAPPbeta) were reduced in the CSF of patients with HIV dementia [81]. Even though the etiology of HAND overlaps with several aspects of Alzheimer’s disease, some studies suggest that chronic HIV infection causes the deposition of amyloid protein in the brain. However, the direct relation of this deposition to the clinical state remains elusive [54,82,83]. Interestingly, the CSF expression pattern of Aβ1-42 fragment and total tau (t-tau) is similar in patients with Alzheimer’s disease and those with HIV dementia [84,85,86,87].

In addition to Aβ and tau proteins, CSF neurofilament light chain (NFL), a component of myelinated axons, has also recently been identified as a sensitive biomarker of neuronal injury in HIV infection and multiple other neurodegenerative diseases [88,89,90,91]. During HIV infection, the CSF NFL levels are increased in patients with and without HAND; however, the higher levels in HAND suggest severe neuronal injury [88,89]. A more specific CSF biochemical marker (s) that distinguishes HAND from AD or other neurodegenerative diseases would thus be of considerable clinical importance.

## 8. Plasma

Biomarkers in plasma or CSF that can predict the onset of HAND disease progression would be valuable in the study of treatment responses [66,92]. Plasma biomarkers of HAND include Aβ, tau, plasma albumin ratios, and NFL [92,93,94]. Among these, NFL is undetectable in mild HAND or in virally suppressed individuals. Other plasma markers, such as tau and albumin ratios, are a reflection of BBB integrity and, remarkably, the degree of BBB leakage correlates with the severity of HAND [91].

The study of extracellular vesicles (exosomes) and plasma microRNAs (miRNAs) is an emerging area of interest, specifically in the context of HIV-1 and other diseases. Exosomes mediate disease progression by acting as conduits that communicate with other cells via their specific cargo in both the CNS and other tissues [95]. These vesicles carry cargo comprising cellular proteins, DNA, lipids, mRNA transcripts, and miRNAs. Recently, the exosome field has garnered immense interest in the areas of HIV-1 and HAND. Exosomes are generated by most cell types in the CNS, such as neurons, astrocytes, oligodendrocytes, and microglia [66,95]. These extracellular vesicles are secreted by neural cells under both normal and pathological conditions. They have been detected in various body fluids, including, but not limited to, the CSF [96] and plasma [97,98,99], as well as in the adult human brain [96]. Studies by the Pulliam lab have reported several biomarkers of neuron-derived exosomes in the plasma, including CD63 and CD81, with elevation of these markers suggestive of cognitive impairment [100].

Non-coding (nc) RNA molecules comprising microRNAs and long noncoding RNAs have been implicated as important regulators of both the host and viral gene expression, with a contribution to HAND pathogenesis, and also serve as valuable diagnostic biomarkers [101]. Many miRNAs have been reported in the brains of HIV-affected individuals with HAND, non-HAND, and blood compartments of HIV-infected elite controllers and viremic patients [102,103]. Some of these miRNAs are specific to HAND, such as miR125b and 146a, which play a key role in microglial infection and cell death [104]. In another study, these microRNAs were also found to be associated with HIV-encephalitis (HIVE) [104].

Based on the key roles of miRNAs in HAND disease pathogenesis, exosomes carrying cellular miRNAs could thus be utilized as potential biomarkers for HIV infection and HAND and other neurocognitive diseases [101]. The exosomes and miRNAs are stable and can be detected by sensitive assays. More research is warranted to assess patient plasma and CSF exosomes derived from the CNS in the context of HAND disease severity.

## 9. Alzheimer’s Pathology in Animal Models of HAND

Research attempts rely upon relevant animal models to study how systemic infection and immune activation drive neuronal damage as it is difficult to study neuropathogenesis in living patients. Rodent as well as nonhuman primate (NHP) models of HAND have proved valuable for assessing HIV-associated neuropathology. Since HIV is not infectious in rodents, creating effective rodent models that accurately mimic human pathology has been challenging. There is a multitude of factors that need to be considered for a relevant animal model, such as (a) the model should have susceptible target cells; (b) prolonged immune activation is needed to model the chronic nature of the disease; (c) viral load in the target cells as well as CNS needs to be maintained. There are several research and review articles that have extensively studied animal models of HAND, which are summarized in Table 1.

HIV-mediated chronic inflammation and immune activation could pave the way for neurodegenerative disorders. Interestingly, HAND patients also exhibit features that are very similar to early and late stages of AD [67,126]. Emerging studies suggest that HIV infection directly or indirectly disrupts several steps in the synthesis and clearance pathways of Aβ and Tau proteins by various mechanisms [127]. Furthermore, there is considerable evidence for increased Aβ accumulation in HIV brains compared with age-matched uninfected controls. Interestingly, however, the Aβ accumulation patterns in HIV brains were found to be distinct from those in AD brains [82,83]. Based on the extensive literature, the ideal small animal model for studying the neuropathogenic mechanisms leading to the unique pattern of Aβ deposition in HIV brain is the gp120 transgenic mice model. Studies have shown that gp120 Tg mice display several features of AD, such as the accumulation of increased levels of proneurotrophin brain-derived neurotrophic factor in the hippocampus with decreased dendritic spines [128], impaired neurogenesis [129], impairment in sensory-motor gating [130], and cognitive deficits [131], thus suggesting that gp120 Tg mice are a suitable model for HAND in the context of AD.

HIV-1 is known to alter several steps in the metabolism of Aβ [126,127]. HIV-1 proteins, e.g., Tat, gp120, Vpr, Nef, and the fusion Gag-polymerase polyprotein (Gag-Pol), have been shown to affect amyloid processing, aggregation, and/or subcellular localization [132,133]. Although some of the viral proteins likely affect Aβ metabolism indirectly, other viral factors appear to function more directly via different mechanisms. For instance, gp120 and Nef can cause infected microglia or astrocytes, respectively, to release pro-inflammatory cytokines, which, in turn, increases the accumulation of Aβ, whereas Tat can directly modulate the trafficking and processing of the amyloid precursor proteins (APP) involving inhibition of the Aβ-degrading enzyme neprilysin [126,134]. Tat is also known to reduce the clearance of amyloids from the brain to the blood and promote the nuclear entry of Aβ and inflammatory responses in endothelial cells in the human brain [127]. Tat, however, is secretory in nature and is highly toxic, thereby suggesting that some of its effects on Aβ production could be indirect [135].

Recent findings have identified viral proteins that induce Aβ production and the mechanisms associated with it. Although APP has few biological functions, it was recently found to act as an innate restriction factor that inhibited HIV-1 release from brain-resident microglia as well as macrophages. To evade this restriction, HIV-1 Gag polyprotein binds and directs the secretase-driven processing and degradation of APP, resulting in amyloid production, leading, in turn, to neurotoxicity [136]. Supporting this notion, another recent report suggested a role of β-secretase BACE1 and Aβ oligomer production in HIV-associated neurotoxicity [137]. Because of this, Aβ production, in part, is a detrimental side effect of a viral evasion mechanism to escape restriction to late-stage infection mediated by APP, but it results in neurotoxic effects. Moreover, the virus-specific activities of Gag-induced APP processing, as well as the effects of various other viral proteins on amyloid metabolism, likely contribute to the underlying differences in the patterns of amyloid deposition in the brains of patients with HAND versus AD. Gp120 has been shown to increase APP levels [138], whereas APP can protect against gp120-induced brain damage [139]. In addition, Aβ has been proposed to exert antimicrobial activities as well [140]. Combined with inflammatory responses with infection and Tat cytotoxicity, Gag-mediated APP processing, and Aβ production are likely important contributing factors to the overall complexity of HAND pathogenesis. Furthermore, Gag-mediated induction of APP processing does not require protease activity of the Gag-Pol fusion polyprotein, which was reported to cleave APP [141]. Instead, Gag binds to APP and enhances its processing by host secretases [136]. Inhibiting either β- or γ-secretase suppresses Gag-mediated amyloid production and protects against neurotoxicity [136]. By protecting APP from degradation, secretase inhibitors were also shown to suppress HIV-1 release and replication in microglia and macrophages, suggesting thereby that these inhibitors could have the potential to serve a dual purpose of preventing both neurodegeneration and HIV-1 replication in brain-resident target cells [136]. Although the role of APP and Aβ in HAND remains complex and somewhat controversial, similar to that with AD, these recent findings provide additional evidence that infected macrophages and/or microglia produce toxic amyloids and could, at least, in part, explain why and how infection causes this induction. There is undoubtedly more to learn about the molecular basis by which this phenomenon occurs, as well as the extent to which Aβ contributes to neurodegenerative disease in affected patients.

## 10. Role of Different CNS Cell Types in Alzheimer’s-Like Pathology

### 10.1. Neurons

Neurons are key players in the CNS. Their physiology and charged nature allow these cells to connect and communicate with other neurons through synapse development. Synapses ensure that organisms process and respond to stimuli in a functional manner; thus, the degradation of synapses results in deficient motor function, memory, speech, appetite, and many more essential processes [142].

As previously discussed, PLWH show higher rates of dementia and accelerated aging processes relative to healthy individuals [143]. Pharmacological treatments combat the lethal frank dementia that previously plagued HIV patients, but the pattern of neurocognitive dysfunction, albeit with varying severity, continues even in the era of cART [144]. HIV does not directly infect neurons; however, it does infect peripheral cells such as monocytes and macrophages, which, in turn, can cross the BBB and can trigger the release of inflammatory factors that degrade cells, including neurons [145]. In recent years, post-mortem studies have also shown higher than average deposition of Aβ in the brains of HIV patients [7,54]. Furthermore, this pathology is associated with the dysregulation of genes and proteins associated with both AD and degraded neuronal integrity [82,146].

If this pattern of overlap between HAND and AD extends to cultured neurons, the capacity for identifying mechanism (s), classifying cell-type-specific involvement, and developing therapeutic intervention broadens. Evidence shows that this overlap does exist. Given the fact that individuals on cART maintain high HIV protein viral loads in the CNS throughout their lifetime, HIV proteins are potential causes for the degeneration and amyloid pathology seen in HAND neurons. The HIV Tat protein is one such protein that persists in the CNS despite cART therapy [147].

HIV-1 Tat has been shown to bind with and disrupt the normal function of multiple proteins associated with the breakdown and production of amyloids in the brain. Endolysosomes are organelles that express BACE1 and whose disruption is associated with the early stages of AD [148]. Exposure of primary human neurons to Tat results in changes to the structure of the endolysosome and also increases its Aβ and APP cargo [149]. Additionally, after noting the time- and temperature-dependent nature of Tat’s effect on neuronal degradation in vitro, one group found binding of Tat with the lipoprotein receptor-related protein (LRP), using a yeast two-hybrid cloning assay. This group deciphered the specific domains of Tat that bind LRP and found that blocking this interaction reduced the neuronal uptake of exogenous Tat [150]. Having shown this interaction, the group wished to understand its effect on the homeostatic function of LRP, a protein that normally binds, endocytoses, and catalyzes the breakdown of AD proteins such as Aβs, apolipoprotein E4, and APP. After a co-incubation period of two hours with Tat, human primary neurons exposed to AD proteins were much less likely to uptake these known LRP ligands [150]. If cells are not endocytosing exogenous amyloids, it is possible that they would accumulate extracellularly and form plaques. While blockage of LRP is a compelling reason for the lessened uptake, Tat also inhibits neprilysin (NEP), a membrane bound Aβ-degrading enzyme, in brain aggregates in vitro [151]. Using a similar test to the previously mentioned preliminary co-incubation assay, brain aggregate membrane fractions were found to display a Tat-concentration-dependent inhibition of NEP activity [151]. Furthermore, compared to known NEP inhibitor thiorphan, Tat is more effective at increasing Aβs in these brain aggregates [151]. Both studies confirm their findings in post-mortem macaque tissue and human brain tissue, respectively, and thus present a convincing argument that the presence of HIV proteins indirectly leads to the accumulation of amyloids presented by the AD phenotype.

There is also an interplay between dysregulated lipids and cholesterols in Aβ aggregates and with Tat. Inhibition of cholesterol and lipid production has been shown to reduce Aβ accumulation, stimulate APP degradation, and predict AD pathology in multiple studies [152,153]. Tat and cocaine together have been shown to upregulate genes involved in cholesterol transport and metabolism, as well as the expression of the proteins associated with these processes in neurons [154]. Extracellular cholesterol is also increased in response to this treatment, a factor potentially contributing to plaques [154]. Moreover, high abundance of lipid rafts in neurons is normally involved in neuronal damage. It has been reported that HIV Nef reduces the level of ABCA1 (cellular cholesterol transporter), leading to the accumulation of intracellular cholesterol and inducing pathological changes to lipid rafts [155]. Interestingly, Ditiatkovsky et al. have reported that Nef secreted in extracellular vesicles (ExNef) was taken up by neurons, causing a reduction in ABCA1 and the rate of cholesterol efflux. ExNef caused the redistribution of APP and Tau to the lipid rafts of neurons and thereby increased the abundance of these proteins along with Aβ. Misfolding and aggregation of amyloidogenic proteins increase inflammatory signaling, leading to neuronal apoptosis and neurodegeneration [156].

There is also evidence for Tat having a more direct role in Aβ accumulation and degenerative pathology in neurons. In vivo Tat injections in a mouse model of AD resulted in the colocalization of Tat with Aβs, and with APP [67]. One group characterized the mechanisms and functional relevance of this interaction in great detail [67]. Under ThioflavinT bulk and single fibril fluorescence, Tat increased β-sheet formation within the Aβ samples, interacted directly with specific regions of Aβ, and, under high concentrations, changed the nature of these biological fibers [67]. Additionally, these Tat/Aβ complexes showed greater adherence to neurons, and their presence correlated with reduced cell viability and stunted neurite outgrowth in cell culture [67].

There is evidence of both direct and indirect interactions between APP and Tat, likely contributing to the intraneuronal AD phenotype. In a study, it was shown that rat primary cortical neurons treated with supernatants from HIV-infected macrophages demonstrated increased expression of BACE1, the enzyme critical for inappropriate cleavage of amyloid precursor protein (APP). This study also showed a mechanism dependent on NMDAR, a receptor found on glutamatergic neurons [137]. In the previously mentioned study regarding increased Aβ-accumulation in endolysosomes, the authors also noted an increase in BACE1 activity following Tat exposure [149]. Furthermore, in U-87 MG cells that express Tat, there was the localization of Tat with cytosolic APP, which promotes the colocalization of lipid rafts with the Tat–APP complexes [157]. This interaction leads to increased β-secretase activity and increased Aβ aggregates. These findings correlated with those reported earlier, suggesting that, in both the basal ganglia and temporal cortices of SIV-infected macaques with encephalitis, there was colocalization of APP and Tat [150]. Follow-up studies regarding neuronal endocytosis of Tat and the effects on Aβ production are warranted in this field.

While the interactions of Tat and amyloids in neurons have been extensively characterized, there are also suggestions that other HIV proteins that are present in the CNS despite cART also cause neural degeneration via pathways and mechanisms common to AD. For example, both Tat and Gp120 increase the expression of misfolded Aβs and cause neuronal injury in rat hippocampal cultures [158]. Gp120 is an envelope protein of the HIV virus. It has also been shown to produce peroxides via NOX2, which, in turn, dysregulates the cytoskeletal envelope in a manner similar to Aβs [159]. Gp120n also disrupts axonal integrity, leading to increased β-APP immunoreactivity [138]. VPR, another HIV viral protein, causes impaired cognition when expressed by rat astrocytes in vivo. Further investigation regarding its effects on intraneuronal Aβs is warranted [160,161]. Overall, numerous HIV proteins have neurodegenerative effects, and investigating their overlaps with the AD phenotype could lead to a deeper understanding of the detailed molecular mechanisms behind these processes.

The lysosomal cysteine protease cathepsin B, which is elevated in individuals with neurological degeneration, particularly AD, colocalizes with beta amyloids in individuals with HAND and AD [162,163]. One study showed that deletion of the cathepsin B gene in AD mice models not only altered the patterns of Aβ deposition but also improved memory function in vivo [164]. In multiple studies, it has been shown that cathepsin B causes apoptosis and the degeneration of neurons and that HIV infection precipitates this event [163,165]. One group even showed that the administration of cathepsin B antibodies improved neuronal function [163]. Given the intimate relationship between cathepsin B, AD, HAND, and neuronal degeneration, this protease represents an important target for the development of therapeutic strategies aimed at alleviating HIV-related cognitive deficits that are associated with AD pathology.

Finally, HIV infection, HAND, and amyloid pathology are closely associated with neuroinflammation. HIV/AIDS is an autoimmune disease, directly infecting T cells, peripheral and central macrophages, and other glial cells, which release proinflammatory cytokines such as interleukin-1β, interferon-α, and TNFα [166,167,168]. HIV proteins, such as Nef, have also been linked to the downregulation of crucial scavenger proteins such as CD63 [167]. Many of these factors are also components of AD pathogenesis [169]. Our group has studied the effect of neuroinflammation on endothelial cells, pericytes, astrocytes, and microglia, and has correlated these effects with neuronal degeneration. Studying the interplay between various cell types is essential given the effects that the released factors have on the surrounding environment. Models proposing interventions that can block neuroinflammation are of value for the treatment of AD [170]. Future comparative studies assessing AD patients and HIV-positive individuals are needed in order to understand the intersection and overlap between these two diseases and also to better understand how we can alleviate the inflammatory symptoms of these processes. Finally, neuroinflammation is known to affect the permeability of the BBB, allowing increased and destructive T cell migration via the downregulation of integral structure proteins [171]. Examining cells such as pericytes in the context of HIV and AD pathology is, therefore, highly important and is becoming an area of increased investigation.

### 10.2. Pericytes

Pericytes are key components of the neurovasculature and BBB [172]. The vasculature works to keep the brain oxygenated, a task that is somewhat complicated by the BBB, which keeps exogenous material outside the CNS. Thus, the functions of the pericytes and associated cells are critical for the maintenance of the BBB. Pericytes surround the capillaries, expanding their diameter under active neural conditions, and they support the neurovascular unit [173]. Comprising endothelial cells, astrocytes, perivascular macrophages, and surrounding pericytes, the neurovascular unit forms the tight junctions that make up the BBB. In HAND and AD, it is common to find lapsed pericyte function and weakening of the BBB [174]. Findings from our group have shown that cultured pericytes express the platelet-derived growth factor subunit B and exhibit increased migration in response to viral Tat protein, providing evidence for the vulnerability of the BBB in the presence of HIV proteins [175]. Pericytes have been shown to express a number of primary HIV receptors (CD4, CXCR4, and CCR5) and have also been shown to be infected by HIV [176]. It has been shown that when the integrity of pericyte function is compromised, the resulting weakened BBB allows enhanced l transmigration of infected monocytes and macrophages [176,177]. Given the correlation between degenerating pericytes and HIV/AD neuropathologenesis and the role of a compromised BBB in both syndromes, pericytes represent an important avenue for studying AD pathology in the context of HIV infection.

Cell-specific markers for pericytes are altered in disease states; these include platelet-derived growth factors (PDGFβ), cell adhesion molecules such as I-CAM, NG2, CD13, and CD146, among others [178,179,180]. PDGFβ knockout mice show an age-dependent loss of pericytes in vivo. It has been shown that crossbreeding of these mice with APP overexpressing mice resulted in the accelerated accumulation of Aβ accumulation, which was concomitant with increased tau pathology and cognitive impairment, two features usually lacking in APP transgenic mice. These results indicate the importance of pericytes for plaque development and, in turn, for cognitive decline [181]. Furthermore, in primary mouse cultures, reduced numbers of pericytes were found to correlate with reduced expression of the lipoprotein-related receptor (LRP1) [181]. LRPs are known associates of AD. It is noteworthy that in a study regarding LRP interactions with Tat, mentioned above, APP-immunoreactive neurons were more densely located in the perivascular regions and were located near Tat-stained cells in these areas as well [150].

Tat protein affects the BBB in numerous ways. It affects the expression levels of claudin-5, a protein regulated by pericytes that helps to maintain the tightness of the junctions [182,183]. Furthermore, human brain endothelial cells exposed to HIV-Tat demonstrated increased release rates of exosomes loaded with Aβ, which, in turn, were taken up by the pericytes [184]. HIV-Tat also increased the expression of the receptor for advanced glycation end products (RAGE) in pericytes, which transport Aβs to endothelial cells [185]. Dysregulation of this receptor has long been associated with AD pathology and has been shown to allow T cell transmigration across the endothelial layer in AD patients [186]. A caveolae-dependent mechanism has been implicated in the dysregulated expression of the RAGE receptor [187].

Other HIV proteins also affect BBB permeability. For example, the early protein Nef, expressed by astrocytes, has been shown to exacerbate BBB disruption [188]. Nef-transfected microglia have similar effects on the BBB, with alleviation seen via the use of Nef peptides [189]. Details regarding the specific effects of these proteins on pericytes and how these interactions fuel increased or decreased expression of AD-related enzymes and receptors are missing from the literature and warrant further investigation.

### 10.3. Astrocytes

Astrocytes play an important role in maintaining an ideally suited milieu for neuronal functionality, and they are also involved in the progression and outcome of many neuropathological conditions. It has become increasingly evident that astrocytes are significant contributors to HIV-1-associated neurological disorders by modulating the microenvironment in the CNS and by releasing proinflammatory cytokines. Recent studies have shown direct metabolic interactions between neurons and astrocytes, particularly in HAND. These interactions include dysregulated potassium (K^+^) homeostasis, intracellular calcium concentrations, glutamate clearance, and compromised BBB integrity and permeability. Such dysfunctions are amplified via gap junctions, thus impacting surrounding neurons and significantly contributing to the pathogenesis of HIV-1-associated neuropathology [190]. In our recent study, we have identified yet another novel role of astrocytes in Alzheimer’s-like pathology in HAND. Herein, we have demonstrated a new molecular pathway involved in astrocytic amyloidosis in HIV-associated Alzheimer’s-like pathology [55]. We also reported astrocytic amyloidosis in the archival brain tissue of both rhesus macaques chronically infected with simian immunodeficiency virus (SIV) and humans infected with HIV-1. Interestingly, the extent of amyloidosis in humans was found to correlate with the severity of cognitive impairment. Furthermore, dissection of the molecular pathway(s) involved in the Tat-mediated increased expression of amyloids in human primary astrocytes involved upregulated expression of the transcription factor hypoxia-inducible factor (HIF)-1α and its subsequent interaction with long noncoding (lnc) RNA BACE1-AS to form a unique complex. This complex, in turn, was shown to regulate the synthesis and stabilization of BACE 1 (amyloid precursor protein cleaving enzyme) via multiple regulatory mechanisms (transcription, post-transcription, translation, and post-translation), leading to increased expression of the cleaved neurotoxic toxic Aβ1–42 form. This is the first report of HIV-1 Tat-mediated induction of astrocytic amyloidosis involving the HIF-1α-lncRNA BACE1-AS axis, with a potential contribution to the progression of HAND pathogenesis. Understanding how astrocytes are involved in HIV-Tat-induced Alzheimer’s-like pathology, leading to cognitive deficits and unraveling the therapeutic potential of HIF-1α in this process, is a novel concept and could be the basis for the future development of therapeutics aimed at treating the cognitive impairment (s) associated with HAND. Future ramifications of these studies could include the development of therapeutic strategies for HIV-AD as well as other dementias involving astrocytes. Specifically, since HIV+ individuals are also afflicted with substance use disorders, HIF-1α-based therapeutics could also be envisioned as treatment options for the comorbidity of substance use disorders in HIV-affected individuals.

### 10.4. Microglia

Microglia are the primary cells infected by HIV-1 in the CNS, and they play a key role in the neurotoxicity observed in HAND. Activation of microglia leads to downstream activation of a multi protein complex, the NLRP3 inflammasome [191], and the release of pro-inflammatory mediators and the altered secretion of cytokines, chemokines, secondary messengers, and reactive oxygen species (ROS), which, in turn, can activate signaling pathways that initiate neuroinflammation. ROS and inflammation, in turn, are known to play critical roles in HAND. However, there is a need to study the physiology of microglia and the processes involved in their activation in order to better understand how HIV-1-infected microglia play a role in the development of HAND [192]. A recent study has demonstrated a unique role of microglia in the accumulation of neurotoxic amyloids, which can also contribute to HAND. This study showed that HIV-1 Tat protein inhibited the microglial uptake of Aβ1-42 peptide, a process that was enhanced by interferon-gamma (IFN-γ) and inhibited by the STAT1 inhibitor (-)-epigallocatechin-3-gallate (EGCG). This process promoted a switch from a phagocytic to an antigen-presenting phenotype of microglia through activation of class II trans-activator (CIITA). Additionally, HIV-1 Tat was shown to significantly disrupt apolipoprotein-3 (Apo-E3)-mediated microglial Aβ uptake. As Tat has been shown to directly interact with the low-density lipoprotein (LRP) receptor and thus inhibit the uptake of its ligands, including apolipoprotein E4 (Apo-E4) and Aβ peptide in neurons, a similar inhibition of LRP was also observed in microglia. In summary, this study showed that HIV-Tat disrupted the microglial phagocytosis of Aβ and Apo-E3 via IFN-γ-mediated STAT1 activation [193]. Tat has been shown to inhibit the activity of Aβ-degrading enzyme neprilysin (Aβ-degrading enzyme) in post-mortem brain tissue from patients diagnosed with HIV. Interestingly, recombinant Tat (specifically with the cysteine-rich domain), when added to brain cultures, resulted in a 125% increase in soluble Aβ, and this was due to the inhibition of neprilysin activity [151]. Tat has also been reported to decrease the activity of neprilysin, resulting in the inhibition of the microglial phagocytosis of amyloids and, thus, further contributing to the burden of amyloid load in the brain [193]. Additionally, amyloids can also physically bind with Tat protein to aggravate neurotoxicity in the brains of virally suppressed HIV-affected individuals. Interestingly, recent reports have also shown that antiretrovirals such as zidovudine, lamivudine, indinavir, and abacavir, alone and in combination, can inhibit the microglial phagocytosis of amyloids as well as induce neuronal amyloid production, thereby also contributing to the amyloid burden [12]. FutFure studies are required to assess the involvement of HIV infection, other viral proteins, as well as antiretrovirals in the phagocytic role of microglia in amyloid clearance. Since microglia are known to be long-lasting reservoirs of HIV-1, clearance of these infected cells could be a strategy for ameliorating HAND and attaining a functional HIV cure. Currently, several strategies are being developed to clear the infected cells; these include pharmacological-based approaches of shock and kill, block and lock strategies, and, more recently, CRISPR/Cas9-based gene editing [194,195]. The shock and kill strategy uses latency-reversing agents (LRAs) such as disulfiram and ixazomid as well as T cell activating agents such as prostratin, bryostatin, and ingenol B to reactivate the latent virus [196]. Apart from these, several latency-promoting agents such as Didehydro-corticostatin, an inhibitor of Tat, ABX464, an inhibitor of Rev, and several other miRNAs have also been implemented as inhibitors of HIV-1 gene expression [197]. Despite these efforts, strategies such as shock and kill and block and lock have not been ideal for the removal of infected microglia since reactivation of these cells often leads to neuroinflammation and enhanced HAND pathogenesis. Notably, CRISPR/Cas9 technology has proven to be very promising in this regard, as it operates on specific gene sequences to excise and eliminate the virus from its host genome. As an example, CRISPR/Cas9-based deletion of glia maturation factor (GMF) was found to significantly reduce oxidative-stress-mediated neurodegeneration. Elimination of GMF in microglia also reduced the Nrf2-translocation-mediated activation of heme oxygenase-1/ferritin and resulted in improved mitochondrial dynamics [198]. This gene-editing technology has also been used to produce novel in vitro models of HAND. For example, Champbell et al. have generated a modified HIV provirus by replacing the Gag-Pol region with NanoLuciferase (NanoLuc) using the CRISPR/Cas9 technology to model HIV proviral activity in microglia [199]. Since no effective therapies are currently available for HAND, targeting microglia in this non-cell-specific manner could likely shed light on the neuroimmune pharmacological basis of the disease.

## 11. Conclusions and Future Perspectives

In the cART era, while the incidence of a severe form of cognitive impairment—HIV-associated dementia—has decreased, milder forms of HAND remain an important clinical problem. The survival of HIV patients is on par with that of those not infected with the virus. It is reported that patients over 50 years of age constitute a large population of the HAND cohort. It is well known that age represents a significant risk factor for AD. The question then arises as to whether there is a relationship between HAND and AD. While, in HAND, the causal role of HIV is well known, in AD, to date, there are reports of many microorganisms and other contributing agents as possible factors inducing AD. There are, however, conflicting reports about the deposition of fibrillar brain amyloids in individuals with HAND, as assessed by MRI using Pittsburgh Compound B. In this study, the authors found no correlation between low levels of CSF Aβ 42 and Pittsburgh Compound B mean cortical binding potential (MCBP) in individuals diagnosed with HIV. On the other hand, patients diagnosed with AD with symptomatology exhibited Pittsburgh Compound B-positive plaques. A possible explanation for this discrepancy could be attributed to several factors, including the age of the patients, sample size, region of the brain assessed, and collection methods, as mentioned by the authors themselves, while many other studies report amyloid deposition in the brains of HAND patients.

Although both AD and HAND brains show deposition of plaques, there are marked differences in plaque deposition in HAND and AD patients. AD has been characterized by the deposition of senile plaques, while diffused neuritic plaques are observed in HAND. Interestingly, soluble amyloids either remain unaltered or are increased in AD, while in HAND patients, soluble amyloids are decreased. Recent data show that HIV induces Alzheimer’s-like pathology by increasing Aβ production in virally suppressed HIV patients and in an animal model of HIV. Interestingly, several viral proteins (Tat, Gp120, Nef, and Vpr), which are released from infected cells in the nervous system, cause synaptic injury and pathogenesis of AD. Additionally, newer findings have shown that both astrocytes and pericytes can contribute significantly to the process of amyloidosis. Interestingly, microglia have also been shown to play a role in amyloid accumulation following exposure to Tat. These studies demonstrated that non-neuronal CNS cells such as astrocytes could also play important roles in inducing Alzheimer’s-like pathology in HAND (Figure 2). This area of research needs to be further investigated to decipher the extent of the contribution by these other cells, leading to exacerbated disease pathogenesis. Furthermore, amyloids have been shown to physically bind with Tat protein, resulting in exacerbated neurotoxicity in the brains of virally suppressed HIV-affected individuals. Interestingly, recent reports have also shown that antiretrovirals alone or in combination can inhibit the microglial phagocytosis of amyloids and induce neuronal amyloid production, thereby further contributing to the amyloid burden. Furthermore, AD-associated factors such as BBB regulators, members of the stress-related pathways, as well as the amyloid and Tau pathways, appear to also augment both amyloid plaque deposition and neurofibrillary tangle accumulation following HIV neuroinfection. Recent research advances have shown the lack of animal models that precisely mimic the human disease pathology of AD. However, mouse models with humanized sequences and mutations could overcome these limitations, at least in part. The generation of a knock-in mouse model with a mutated APP gene showed Aβ pathology, neuroinflammation, and memory impairment [1,200]. Similarly, the transplantation of human pluripotent stem cell-derived cortical neuronal precursors into the brains of a murine model revealed AD-like pathological features [201]. Recently, a research group developed a humanized mouse (Human interleukin-34) possessing the human immune system and human glial cells that mimic the innate immune activities of the CNS. The microglia-like cells expressed all major cell markers and receptors that were readily infected by HIV-1 [202]. Though these models are good templates to study neuroinflammation and AD-like pathology, the differences in the immune responses of the human AD brain necessitate the development of rodent models with a complete immune system. Additionally, the involvement of the HPA axis in HIV was shown to be associated with BBB disruption, neuroinflammation, oxidative stress, excitotoxicity, and increased Aβ burden. This, combined with other factors (environmental/genetic), could provide new insights for understanding the pathogenesis, diagnosis, and therapeutics of brain disorders, including Alzheimer’s-like pathology in HAND. The scenario of replication-independent production of HIV-1 protein is apparently counterintuitive, and the underlying molecular mechanisms remain largely unexplored. It is, therefore, critical to explore the role of viral proteins as well as antiretrovirals in Alzheimer’s-like pathology in HAND.

Based on the urgent need for the development of effective therapeutics for HAND, research in the areas of biomarker discovery, extracellular vesicles carrying RNA therapeutics, and agents blocking neuroinflammation should be encouraged. Since amyloids can also serve as a biomarker of the ongoing neuronal injury in the brain, further studies in this respect are required for developing adjunctive therapeutics for HAND. Agents that regulate inflammatory and/or cell death pathways and that mediate neuronal damage such as the formation of toxic amyloid isoforms could thus serve as effective therapeutic targets, ameliorating both synaptodendritic injury and neuroinflammation in HAND.

## Figures and Tables

**Figure 1 vaccines-09-00930-f001:**
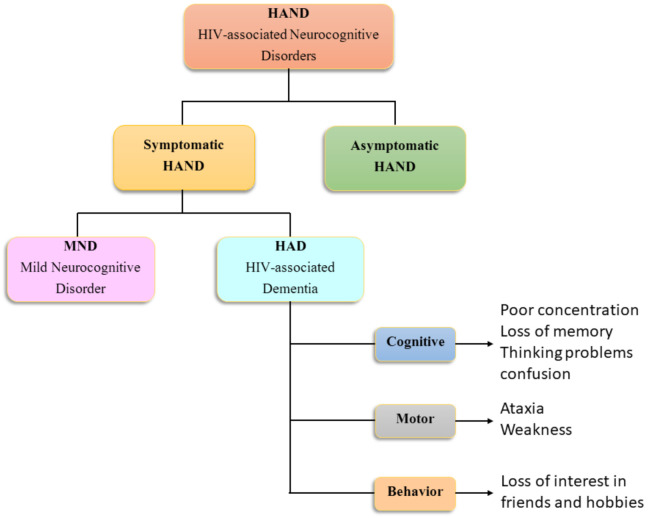
Clinical symptomology of HAND.

**Figure 2 vaccines-09-00930-f002:**
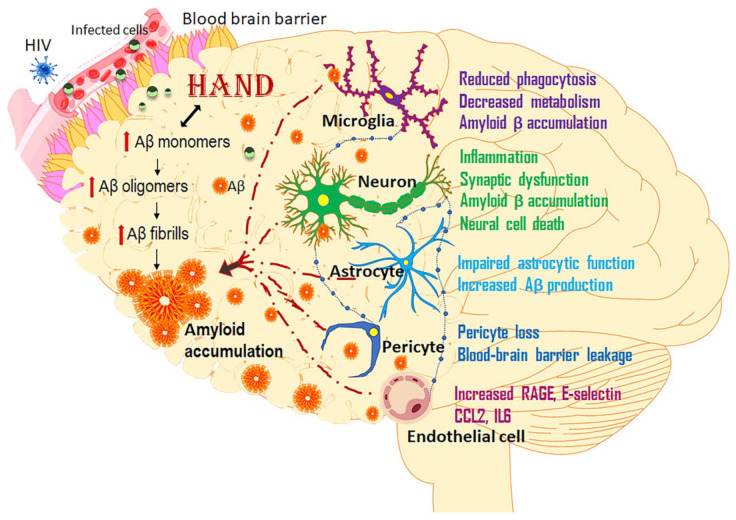
Role of central nervous system cell types in HIV-mediated accumulation of amyloids: both HIV and HIV-infected cells cross the blood–brain barrier and infect glial cells in the central nervous system. Activated neurons, microglia, astrocytes, pericytes, and endothelial cells play a key role in the accumulation of amyloids. Prolonged accumulation of amyloids, in turn, promotes neurodegeneration.

**Table 1 vaccines-09-00930-t001:** Animal models of HAND.

Name	Neuropathogenesis	Reference
***Humanized mice model***		
NOD/SCID mice	Neuronal apoptosis, astrogliosis, microglial activation, neuronal injury	[105,106]
NSG mice	Neuroinflammation, multilineage hematopoiesis, loss of microtubule-associated protein 2, synaptophysin, and neurofilament antigens	[107,108,109,110]
hu-PBMC	Astrocytosis, microglial nodules, upregulation of TRAIL	[111,112]
hu-PBL	Inflammation, immune activation	[113]
BLT	Increased levels of immunomodulatory cytokines and chemokines, Gag-specific T cell responses	[114]
HIVE	Neuronal cell death, microglial activation	[115]
***Transgenic mice model***		
Gp-120	Apoptosis, astrogliosis	[116]
Vpr	Inflammation, neurodegeneration	[117]
GFAP-Tat	Astrocytosis, degeneration of neuronal dendrites, neuronal apoptosis, increased infiltration of immune cells	[118]
Gag-pol deleted HIV-1 Tg	Endothelial apoptosis, reactive gliosis	[119]
***Non-human primate model***		
SIV/17E-Fr Macaques	Encephalitis, immune activation, inflammation	[120,121]
SIVR71/17E Macaque	Microglial inflammation, behavioral deficit, amyloidosis, metabolic changes, microbial dysbiosis	[55,122,123,124]
SIV251 Macaque	Increased microglia/reservoir	[125]

NOD: nonbase diabetic; NSG: NOD scid gamma; SCID: severe combined immunodeficiency; PBMC: peripheral blood mononuclear cells; PBL: peripheral blood lymphocyte; TRAIL: TNF-related apoptosis-inducing ligand; BLT: bone marrow–liver–thymus; HIVE: human HIV-1 encephalitis; GP; envelope glycoprotein; Vpr: HIV viral protein r; GFAP: glial fibrillary acidic protein; TAT: trans-activator of transcription; Gag: group antigens; SIV: simian immunodeficiency virus.

## Data Availability

Not applicable.

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
