# Peer review of "Alzheimer’s-Like Pathology at the Crossroads of HIV-Associated Neurological Disorders"

_vaccines, 2021, doi:10.3390/vaccines9080930_

Round 1
Reviewer 1 Report
This is a rather comprehensive review that focuses on the causal relationship of Alzheimer's-like pathology in HIV-associated neurocognitive disorders, and the contribution of several CNS cell types to this process. There are certain minor issues that have to be addressed before this manuscript can be considered for publication in the journal Vaccines.
First and foremost, there are smaller typographical and language issues throughout the manuscript, which should be amended. In the Abstract section of the manuscript, all abbreviations should be initially stated in full.
The impact of HIV central nervous system persistence on pathogenesis should be tackled in more depth. Please consult: Brew & Barnes. AIDS. 2019;33 Suppl 2:S113-S121. doi: 10.1097/QAD.0000000000002251.
Selective targeting of microglial cell signaling may be a valid option to control HIV-associated neurocognitive disorders (HAND) with lesser side effects, and there is a viable role of CRISPR-Cas9 system that is not mentioned in the manuscript. Please consult: Pahan K. A Broad Application of CRISPR Cas9 in Infectious, Inflammatory and Neurodegenerative Diseases. J Neuroimmune Pharmacol. 2019;14(4):534-536. doi: 10.1007/s11481-019-09889-4.
In the Table 1 not all abbreviations are delineated in full in the table legend; this should be amended.
Finally, the link of this manuscript to the overall topic of vaccines and vaccinology is not clear, but I leave this to editors to decide.
Author Response
This is a rather comprehensive review that focuses on the causal relationship of Alzheimer's-like pathology in HIV-associated neurocognitive disorders, and the contribution of several CNS cell types to this process. There are certain minor issues that have to be addressed before this manuscript can be considered for publication in the journal Vaccines.
First and foremost, there are smaller typographical and language issues throughout the manuscript, which should be amended. In the Abstract section of the manuscript, all abbreviations should be initially stated in full.
Response: Thanks for the suggestion. The abstract and the text has been modified in the revised manuscript and typos taken care of as well.
The impact of HIV central nervous system persistence on pathogenesis should be tackled in more depth. Please consult: Brew & Barnes. AIDS. 2019;33 Suppl 2:S113-S121. doi: 10.1097/QAD.0000000000002251.
Response: Good point. Done !
Selective targeting of microglial cell signaling may be a valid option to control HIV-associated neurocognitive disorders (HAND) with lesser side effects, and there is a viable role of CRISPR-Cas9 system that is not mentioned in the manuscript. Please consult: Pahan K. A Broad Application of CRISPR Cas9 in Infectious, Inflammatory and Neurodegenerative Diseases. J Neuroimmune Pharmacol. 2019;14(4):534-536. doi: 10.1007/s11481-019-09889-4.
Response: Again, a very good suggestion. Done!
In the Table 1 not all abbreviations are delineated in full in the table legend; this should be amended.
Response: All the abbreviations in Table 1 have been delineated in the revised manuscript.
Reviewer 2 Report
The paper is well written and describes AD pathology linked with HIV and lack of antiviral therapy lead to encaphelitis with CD4 cells entering into the brain. Imaging (PAT, MRI) of the brain was done and amyloid depsoition and neuroinflammation seen.
Role of different CNS cell types in Alzheimer’s-like pathology (eg neurons, microglia) in the disease described as well. as well as of peryocytespaper conclusions:
This area 667 needs to be further investigated to decipher the extent of contribution by these other 668 cells. Furthermore, amyloids have been shown to physically bind with Tat protein result-669 ing in exacerbated neurotoxicity in the brains of virally suppressed HIV-infected individual-670 uals.
author contribution acknowledgment and conflict of interest given
it is written as a review paper.
the paper topic is of interest. there are no major spelling errors.
there are one table animal models) plasma markers were also described
as no figures or are present in the paper though and the list of references seems complete. also, no methods are given (and cohort description), pls add if relevant.
I recommend adding at least one figure (could be a descriptive diagram)
Author Response
The paper is well written and describes AD pathology linked with HIV and lack of antiviral therapy lead to encephalitis with CD4 cells entering into the brain. Imaging (PAT, MRI) of the brain was done and amyloid deposition and neuroinflammation seen.
Role of different CNS cell types in Alzheimer’s-like pathology (eg neurons, microglia) in the disease described as well. as well as of pericytes in paper conclusions:
Response: Thanks much for the constructive comments.
I recommend adding at least one figure (could be a descriptive diagram).
Response: We have now added a figure (Fig. 2) as per the reviewer’s the suggestion.
